# Scalable carbon dioxide electroreduction coupled to carbonylation chemistry

Mikkel T. Jensen[1], Magnus H. Rønne [1], Anne K. Ravn [1], René W. Juhl[1], Dennis U. Nielsen [1], Xin-Ming Hu[1], Steen U. Pedersen[1], Kim Daasbjerg[1] & Troels Skrydstrup [1]

Significant efforts have been devoted over the last few years to develop efficient molecular electrocatalysts for the electrochemical reduction of carbon dioxide to carbon monoxide, the latter being an industrially important feedstock for the synthesis of bulk and fine chemicals. Whereas these efforts primarily focus on this formal oxygen abstraction step, there are no reports on the exploitation of the chemistry for scalable applications in carbonylation reactions. Here we describe the design and application of an inexpensive and user-friendly electrochemical set-up combined with the two-chamber technology for performing Pd-catalysed carbonylation reactions including amino- and alkoxycarbonylations, as well as carbonylative Sonogashira and Suzuki couplings with near stoichiometric carbon monoxide. The combined two-reaction process allows for milligram to gram synthesis of pharmaceutically relevant compounds. Moreover, this technology can be adapted to the use of atmospheric carbon dioxide.

[1] Carbon Dioxide Activation Center (CADIAC), Interdisciplinary Nanoscience Center, Department of Chemistry, Aarhus University, Gustav Wieds Vej 14, 8000 Aarhus C, Denmark. Mikkel T. Jensen, Magnus H. Rønne and Anne K. Ravn contributed equally to this work. Correspondence and requests for materials should be addressed to K.D. (email: kdaa@chem.au.dk) or to T.S. (email: ts@chem.au.dk)

Developing efficient electrochemical methods for the conversion of carbon dioxide into small organic C1- and C2-building blocks presents a promising means for the chemical industry to deviate from petrochemical feedstocks[1–6]. Ideally, renewable energy sources such as solar and wind power can be adapted to drive either the electrochemical reduction of $CO_2$ or the production of hydrogen from water electrolysis necessary for the subsequent hydrogenation of $CO_2$, thereby increasing the overall interest and value of such processes. Carbon monoxide represents one of these potential building blocks which formally is produced by the abstraction of oxygen from $CO_2$. This diatomic gas is a principal feedstock for the production of hydrocarbon liquids and bulk oxygenated products such as aldehydes, alcohols, and carboxylates[7–10]. Industrially, CO is produced mainly from either steam reforming or coal gasification, with fossil fuels as the sole carbon source.

As such, considerable efforts by many research groups have been devoted to the identification and study of a wide variety of molecular electrocatalysts for the selective reduction of $CO_2$ to CO. The main challenge is to develop effective catalysts that lower the overpotential necessary for the electron transfer to $CO_2$ to generate the initial radical anion[11–16]. Despite these intensive and successful studies, currently there are no practical and scalable applications of combining the selective electrochemical reduction of $CO_2$ with the synthesis of more elaborate chemicals.

Here we report on the development of an operationally simple and low-cost electrochemical set-up for the production of CO on demand in synthetically useful scales with a commercially available molecular catalyst, iron tetraphenylporphyrin (FeTPP). Particularly valuable is its adaptation to palladium-catalysed carbonylation reactions[17–20] for carbon–carbon and carbon–heteroatom bond formation in the synthesis of pharmaceutically related molecules. Furthermore, we demonstrate the reusability of the electrochemical system and the possibility of exploiting atmospheric $CO_2$ in synthesis.

## Results

**Initial considerations.** Several challenges were encountered in the design of a simple, inexpensive, and operational bench-top set-up for the ex situ generation of CO and its immediate use as a valuable chemical reagent. The two-step protocol developed involves initially an electrochemical reduction of $CO_2$ towards a predetermined amount of CO at a controlled rate followed by Pd (0)-catalysed carbonylation reaction to provide milligram to gram scale quantities of the end product. The development included firstly the identification of a suitable molecular electrocatalyst, displaying sufficient long-term stability in the preparative reduction of $CO_2$ to CO, secondly a simple but high-performing electrode exhibiting the necessary chemoselectivity in the electrolysis process (CO vs. $H_2$ production), thirdly an air-tight reactor system capable of handling two simultaneous reactions, the $CO_2$ electrolysis and the Pd-catalysed carbonylation, and fourthly a small, affordable, and programmable potentiostat/galvanostat to provide a continuous but predetermined amount and rate of CO. Outlined below we describe the systematic investigation that was undertaken to address all the issues.

Among the many different $CO_2$ reduction catalysts reported we chose to focus on the transition metal complex, iron tetraphenylporphyrin, which is commercially available and previously reported to show under electrolysis of $CO_2$ high selectivity for CO production in the presence of Brønsted acids[21]. Other more elaborate iron porphyrins working at lower overpotentials have recently been prepared[22, 23], and these catalysts should be equally adaptable to our set-up. The same is true for the plethora of molecular electrocatalysts based on Co, Ni, Mn, Re

and others[11–16], which have also been demonstrated to promote the desired transformation.

**Preliminary electrochemical investigations.** During our initial investigations of FeTPP with a glassy carbon (GC) electrode it was confirmed by cyclic voltammetry that weak Brønsted acids such as trifluoroethanol (TFE) and phenol could enhance the rate of $CO_2$ reduction significantly (Fig. 1a) as previously reported[24]. Noteworthy, when applying phenol in a preparative one-chamber set-up, deactivation of the anode by polymerisation of the phenol as described in literature took place and decreased the efficiency of the cell[25–27]. To keep the system scalable and maintain low operational costs, various other electrode materials such as stainless steel, silver, and iron were investigated. Encouragingly, with stainless steel electrodes a steady performance for $CO_2$ electrolysis even over several hours was achieved (Supplementary Figs 15 and 20, and Fig. 2e). Furthermore, stainless steel is inexpensive and it easily allows for the fabrication of high surface area electrodes. With a small stainless steel disc electrode it was possible to record a cyclic voltammogram of the FeTPP showing the catalytic reduction of $CO_2$ at a potential starting from –1.5 V vs. Ag/AgCl corresponding to an overpotential of 600 mV (Fig. 1b).

To evaluate the efficiency of the FeTPP electrocatalyst in the conversion of $CO_2$ to CO several experiments were carried out in a two-electrode set-up using stainless steel electrodes as both working and counter electrode (vide infra and Supplementary Table 1). The effect of the applied current was investigated

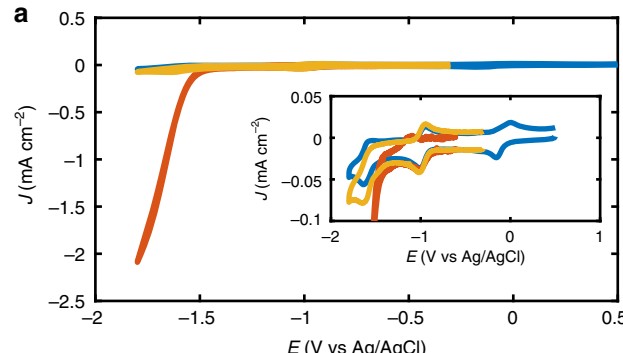

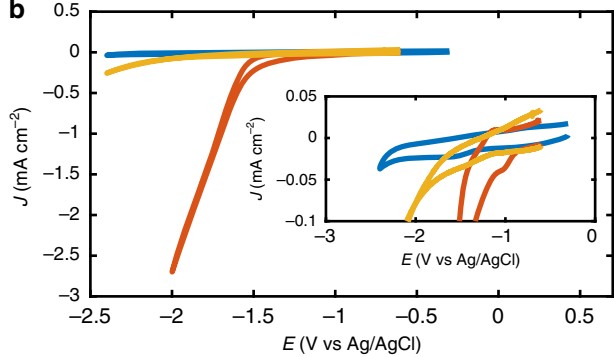

**Fig. 1** Cyclic Voltammograms of FeTPP. **a** Cyclic voltammograms recorded under Ar (*blue*), $CO_2$ (*yellow*) and with 0.8 M TFE added along with $CO_2$ (*red*) at a GC disk electrode (∅ = 0.1 cm; sweep rate = 100 mV s$^{-1}$) in 0.2 mM FeTPP in 0.1 M tetrabutylammonium tetrafluoroborate (TBABF$_4$)/*N,N*-dimethylformamide (DMF). *Inset* shows a blow-up of the voltammograms. **b** Cyclic voltammograms recorded with a stainless steel disk electrode (∅ = 0.3 cm); otherwise same condition as in **a**. *Inset* shows a blow-up of the voltammograms

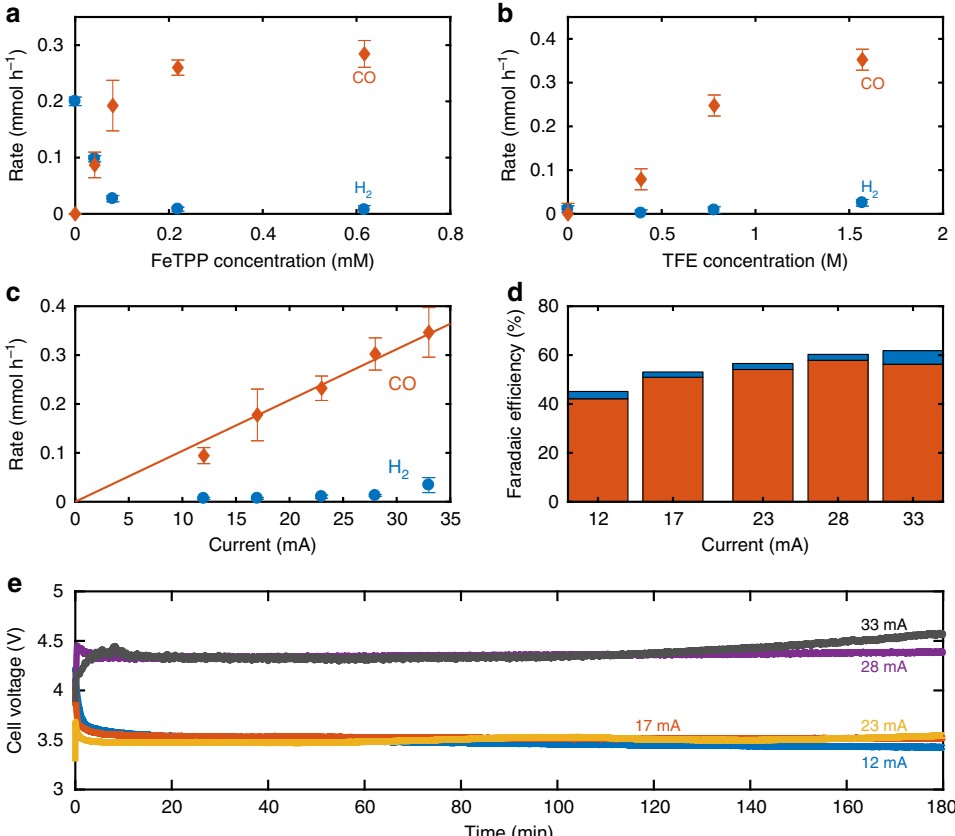

**Fig. 2** Electrochemical reduction of $CO_2$. **a** Rate of CO (*orange*) and $H_2$ (*blue*) produced as a function of FeTPP concentration by applying a constant voltage of 3.8 V at stainless-steel electrodes (Fig. 3a) in 0.1 M TBABF$_4$/DMF containing 0.8 M TFE. **b** Rate of CO (*orange*) and $H_2$ (*blue*) produced as a function of TFE concentration, conditions similar to Fig. 1a but with a fixed FeTPP concentration of 0.2 mM. **c** Rate of CO (*orange*) and $H_2$ (*blue*) produced at various applied currents using 0.8 M TFE and 0.2 mM FeTPP. **d** Faradaic efficiency of the reaction for CO (*orange*) and $H_2$ (*blue*) production, same conditions as in **c**. **e** Cell voltage vs. time for electrolysis using applied currents of 12 (*blue*), 17 (*red*), 23 (*yellow*), 28 (*purple*), and 33 (*green*) mA, same conditions as in **c**. All error bars represent the s.d. for two independent measurements. Gas production was determined by gas chromatography

in a series of galvanostatic experiments (configuration E1; Supplementary Figs 18 and 21). In addition, the stability of the catalyst system was examined along with the effect of varying the concentrations of the catalyst and proton source in two-electrode experiments controlling the voltage (configuration E2; Supplementary Fig. 19). As expected the catalyst concentration is pivotal for controlling the chemoselectivity, in that at least 0.2 mM is required to minimise the competing hydrogen evolution reaction from the TFE (Fig. 2a). Upon adding TFE as proton donor a steady increase in the production of CO occurs. The effect declines going towards the high-concentration range where an undesired increase in the production of hydrogen takes place (Fig. 2b). From this we selected 0.8 M TFE as the optimal condition to balance the positive effect of increased rate with the added material costs related to the use of additional proton donor.

Variation of the applied current in the range from 12 to 33 mA gratifyingly revealed an almost perfect linear relationship with the release rate of CO (Fig. 2c), thereby allowing the rate of CO produced to be adjusted to exactly what is requested by the follow-up reactions that consume CO (vide infra). Unless otherwise noted, 23 mA was used as the standard condition providing good efficiency for CO release with a rate of 0.25 mmol h$^{-1}$. Higher current strengths led to a small contribution of undesirable hydrogen evolution (Fig. 2c) and only a modest increase in the faradaic efficiency of the $CO_2$ reduction from 54 to 57% (Fig. 2d) at the expense of a significant and undesired increase in the

overpotential (Fig. 2e). In general, the Faradaic efficiency for CO production is lower than the state-of-the-art reported for FeTPP (> 90%) in DMF (Supplementary Fig. 22)[21]. This can be attributed to the use herein of the one-chamber electrochemical set-up for electrochemical $CO_2$ reduction with the cathode and anode being placed so close to each other that part of the charge consumed at the cathode goes to the reduction of the oxidised species formed at the anode.

**Electrochemical set-up for bulk synthesis.** To perform bulk electroreduction on a mmol scale of $CO_2$ to CO with subsequent use of the latter in a Pd-catalysed transformation, we adapted an air-tight two-chamber reactor as previously described by us (Figs. 3a, b)[28, 29], though with suitable Teflon and silicon caps for allowing entry of both electrodes. Access of the produced CO into the second chamber for the reaction is possible through the glass bridge. Different sizes of the two chamber reactors were adopted, depending on the scale of the reaction (Supplementary Fig. 14). To further simplify the electrochemical operation, we constructed a simple galvanostat (denoted ElectroWare) capable of delivering either a constant voltage (0–10 V) or current (0–200 mA), while collecting and storing the data on a.csv file (Fig. 2a, Supplementary Figs 16 and 17).

Initial experiments were performed to evaluate the performance of the electrochemical reduction of $CO_2$ adapted to the synthesis of the antidepressant moclobemide (**1**) via a

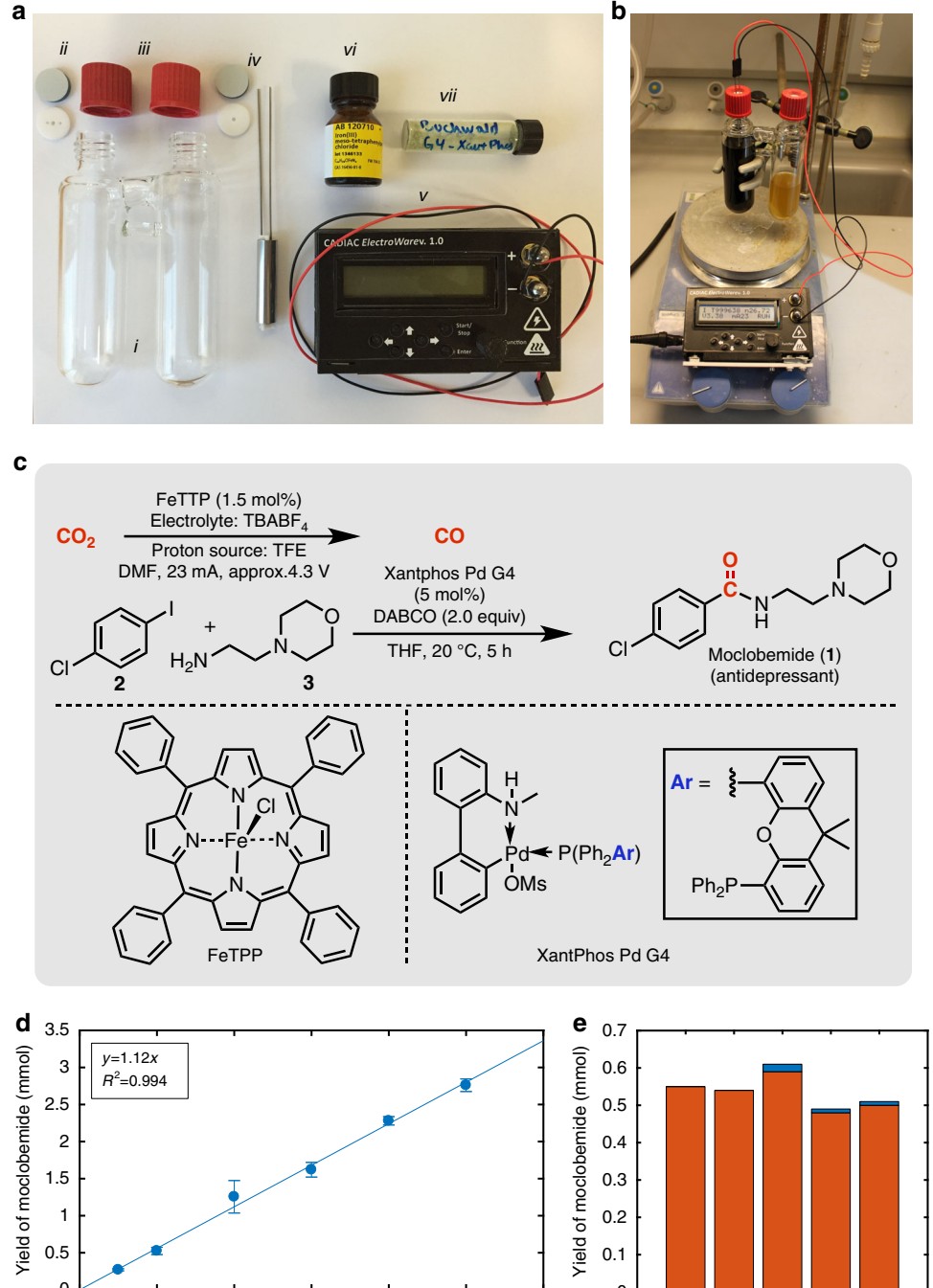

**Fig. 3** Design of electrochemical $CO_2$ reduction equipment, precision studies for CO production and adaptation to a Pd-catalysed carbonylative coupling. **a** The individual parts include (i) a double chamber reactor, (ii) silicon and Teflon seals with appropriate holes for electrodes, (iii) lids for the two chambers, (iv) stainless-steel working electrode (surface area = 24 cm²) and counter electrode (surface area = 7.5 cm²) separated by a Teflon spacer, (v) galvanostat (ElectroWare), (vi) molecular electrocatalyst, FeTPP, for $CO_2$ reduction and (vii) the Buchwald 4th generation Pd(0)-precatalyst (Xantphos Pd G4) for aminocarbonylation. **b** The whole set-up assembled and running. **c** Illustration of reaction conditions for the $CO_2$ reduction to CO and the subsequent aminocarbonylation between aryl iodide **2** and amine **3** for the preparation of moclobemide **1**. **d** Individual reactions for the precision studies were achieved with 3.0 mmol aryl iodide, 6.0 mmol amine, 5 mol% Xantphos Pd G4, and 6.0 mmol DABCO in the chemical reaction chamber, and $CO_2$ saturated 0.1 M TBABF₄/DMF in the electrochemical chamber. Reactions were run with the ElectroWare set to prepare from 0.25, 0.5, 1.0, 1.5, 2.0 to 2.5 mmol CO. The yield of moclobemide produced was determined by high-performance liquid chromatography (HPLC) analysis in comparison to an internal standard (Supplementary Methods). The error bars represent the s.d. for two independent measurements. **e** Demonstration of the reusability of the same set-up as employed in **d** by repeating the reaction sequence and analysis multiple times. The ElectroWare was set to produce 0.5 mmol CO and the yield of the moclobemide (*orange column*) generated was determined by HPLC analysis of an aliquot from the chemical reaction chamber. After each experiment, the electrodes were removed and cleaned. Some minor reduction of 1-chloro-4-iodobenzene (**2**) to chlorobenzene (*blue column*) was observed for runs 3–5

Pd-catalysed aminocarbonylation (Fig. 3c)[30, 31]. The precision of the CO production was evaluated in a series of experiments performed on a 3.0 mmol scale of the aryl iodide **2** and the corresponding amine **3**. In each experiment, the galvanostat was pre-programmed to generate from 0.25, 0.5, 1.0, 1.5, 2.0 to 2.5 mmol CO from the $CO_2$ saturated DMF solution containing the

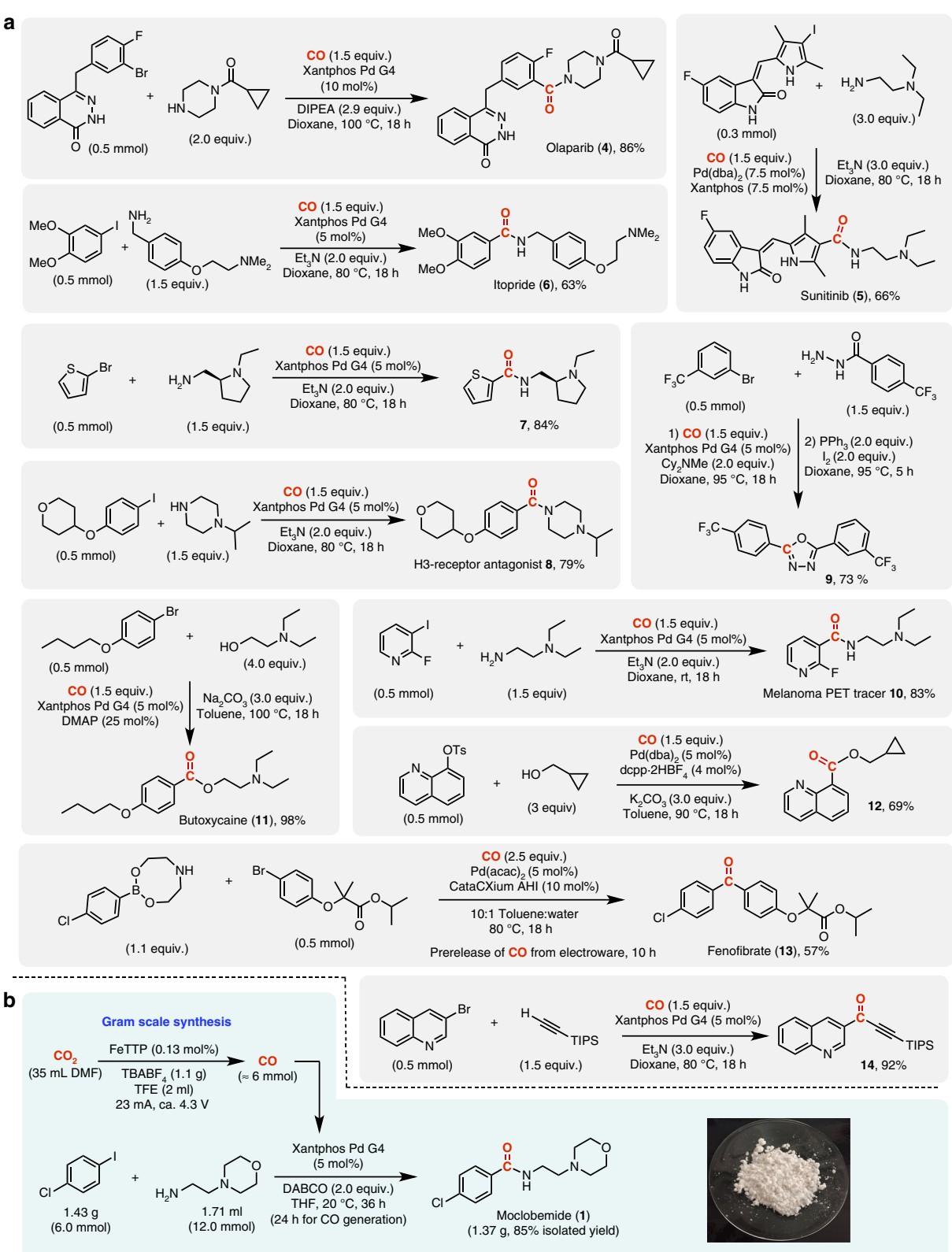

**Fig. 4** Application of the electrochemical $CO_2$-to-CO reduction step to carbonylative couplings. **a** Synthesis of pharmaceuticals and pharmaceutically relevant molecules on a 0.5 mmol scale. All yields refer to isolated yields (average of two runs) after column chromatography on $SiO_2$. **b** Scale-up experiment to the gram synthesis of moclobemide

catalyst FeTTP. The quantity of moclobemide produced was measured directly from the reaction mixture by HPLC analysis compared to an internal standard. As can be seen from the graph (Fig. 3d), a good correlation ($R^2 = 0.994$) with a slope = 1.12 was obtained for the amount of moclobemide (average of two runs) generated with the quantity set for electrochemical CO production. The higher than unity slope would suggest that the amount of CO produced is slightly greater than expected. Presumably, this can be attributed to difficulties in accounting for the pressure increase when determining the amount of CO using gas chromatography.

In a second set of experiments, we investigated the stability of the catalytic system in multiple runs for moclobemide synthesis using 3.0 mmol aryl iodide and 0.5 mmol of CO produced with the ElectroWare. After each run the reaction mixture was analysed by HPLC and the electrodes were cleaned or replaced with a new set. In all, five runs were performed and gratifyingly, in each case, the conversion of $CO_2$ to CO was constant as observed from the production of moclobemide (Fig. 3e), testifying to the stability of the catalytic system for selective $CO_2$ reduction. In runs 3, 4, and 5, a small amount of hydrogen was formed as observed from the reduction of the 1-chloro-4-iodobenzene to chlorobenzene. Omitting the cleaning process results in a decrease of the efficiency (Supplementary Methods and Supplementary Fig. 23).

**Substrate scope**. Next, we explored the suitability of our technology for the preparation of a variety of pharmaceutically relevant molecules employing a Pd-catalysed carbonylation reaction as the key assembling step (Fig. 4 and Supplementary Figs 1–12). These transformations were run on a 0.5 mmol scale with the ElectroWare set to produce ~1.5 equiv. CO from a $CO_2$ saturated 0.1 M TBABF$_4$/DMF solution. Reaction conditions for each of the individual carbonylative couplings were previously determined applying our two-chamber technology with either COgen[30] or SilaCOgen[32] as the CO source.

The active ingredient in three commercially available pharmaceutical drugs could be effectively prepared via an aminocarbonylation with CO produced from the electrochemical reduction of $CO_2$, including the PARP inhibitor, olaparib (**4**, 86% yield), the tyrosine-kinase inhibitor, sunitinib (**5**, 66% yield) and the D$_2$-receptor antagonist, itopride (**6**, 63% yield). Other compounds successfully prepared from an aminocarbonylative step include thiophene carboxamide **7** (84% yield), the H3-receptor antagonist **8** (79% yield), oxadiazole **9** (73% yield) prepared from a dehydrative cyclisation of the intermediate diacyl hydrazide[33], and the melanoma positron emission tomography tracer **10** (83% yield). Resorting to a Pd-catalysed alkoxycarbonylation also proved viable providing the local anesthetic, butoxycaine (**11**) and the cyclopropyl ester **12** in good yields, 98 and 69%, respectively.

Two carbonylative cross coupling reactions for carbon–carbon bond formation were tested as well, including the three component carbonylative Suzuki coupling for the synthesis of the blood cholesterol lowering drug, fenofibrate (**13**), in 57% yield and a carbonylative Sonogashira coupling for the preparation of ynone **14** in 92% isolated yield. Finally, moclobemide synthesis was increased to a 6.0 mmol scale with an approximate 6 mmol production of CO. This led satisfyingly to the gram scale synthesis of this reversible inhibitor of monoamine oxidase A in 85% yield.

**Studies with atmospheric $CO_2$**. In a final study, we examined the possibility of exploiting atmospheric $CO_2$ for the synthesis of **1** on a 0.5 mmol scale with respect to aryl iodide **2**. As such we designed a triple chamber reactor (Fig. 5), to which the first chamber was loaded with the $CO_2$ binder **15** (5.0 mmol) in dimethyl sulfoxide (Supplementary Figs 13 and 24–28 and Supplementary Table 2)[34]. Atmospheric air was bubbled through this solution for six days whereafter heating in the closed chamber to 130 °C allowed for $CO_2$ release. The electrolytic reduction of the released $CO_2$ was set for the production of 1.5 mmol of CO, which after the aminocarbonylation provided moclobemide in an outstanding 99% yield.

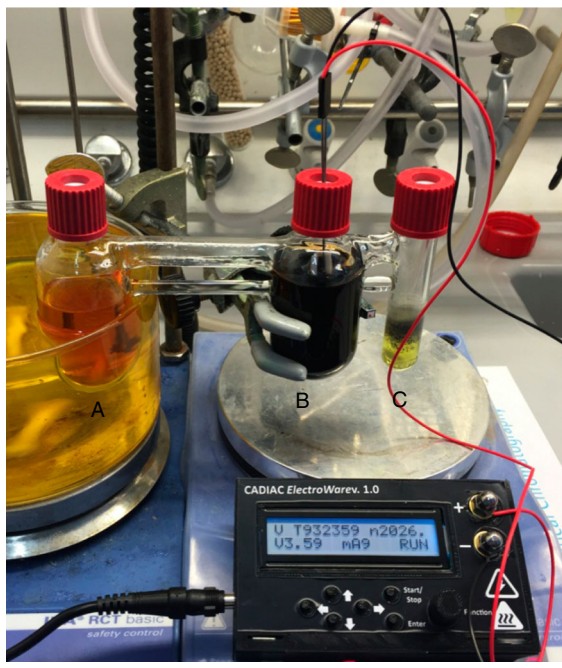

Chamber A: Atmospheric $CO_2$ capture with binder

$CO_2$ binder **15** (5.0 mmol)

Chamber B: Electrochemical $CO_2$ reduction (1.5 mmol CO production)

Chamber C: Moclobemide synthesis on a 0.5 mmol scale

**Fig. 5** Synthesis of moclobemide applying atmospheric $CO_2$. A three-chamber system was adapted including a chamber A containing a $CO_2$ binder **15** (5.0 mmol) in dimethyl sulfoxide, which was bubbled with dry air for 6 days. The second chamber B was set up as above for the production of 1.5 mmol CO employing an applied voltage of 3.6 V. In chamber C, a 0.5 mmol scale aminocarbonylation reaction was set-up including aryl iodide **2** (0.5 mmol) and amine **3** (1.0 mmol) with Xantphos Pd G4 (5 mol%) and DABCO (2 equiv.) in tetrahydrofuran

## Discussion

We have reported on the development of an inexpensive and user-friendly electrochemical device for carrying out the proton-coupled electroreduction of $CO_2$ to CO with an iron porphyrin catalyst, and its subsequent exploitation to four different palladium-catalysed carbonylation reactions, e.g. amino- and alkoxycarbonylations, in addition to carbonylative Sonogashira and Suzuki couplings. This simple set-up allows for the generation of biologically active compounds and simultaneously avoids the concerns of working with CO, which is generated inside the electrochemical reaction chamber and only in near-stoichiometric amounts with respect to the ensuing carbonylation reaction. Particularly noteworthy is, that our process can be exploited for performing carbonylative couplings in milligram to gram scales and that the chemical technology developed exploits a waste greenhouse gas as one of the reagents. Advantageously the costs for performing the electrochemical reduction are low as we have designed and built a simple galvanostat/potentiostat combined with stainless-steel electrodes.

Although our experiments reveal the successful adaptation of electrochemistry with transition metal catalyzed carbonylations for chemical synthesis, there is still room for improvement. For example, the kinetics for the electrochemical $CO_2$ reduction needs to be enhanced in order to embrace other carbonylation reactions as well. So far, our equipment can be set to produce from 0.5 to ~6 mmol CO, but the specific amount of CO generated is slow and time dependent. Hence, 0.5 mmol of CO takes ~2 h, whereas 6 mmol will require up to 24 h. A more rapid production of CO is required if we are to include other carbonylative transformations with stoichiometric CO, such as double carbonylations for α-ketoamide synthesis and carbonylative Heck couplings[35–37], which both require a significant CO partial pressure from the reaction start. As expected these Pd-mediated carbonylations proved to be less efficient with our set-up. Alternatively, other electrode materials or more efficient electrocatalysts could be considered for achieving a faster $CO_2$ to CO conversion[22, 23].

Another adaption of our system would be to identify conditions for performing the $CO_2$ reduction in a low-cost and environmentally friendly solvent, such as water. Recently, it has been demonstrated for iron and cobalt porphyrin molecular catalysts that they exhibit good activity and selectivity for the electrochemical $CO_2$-to-CO conversion in pH neutral water when immobilised on an electrode surface[38–42]. In principle, it is possible to couple the $CO_2$ reduction in water with the carbonylation reaction. However, for the presented system where the electroreduction/oxidation is in one compartment, water based systems should be avoided since water oxidation at the counter electrode leads to the production of $O_2$, which would be detrimental for the Pd-catalysed carbonylative couplings. A solution could be found by using a sacrificial anode or by separating the counter electrode from the working electrode, but this requires a third compartment and thereby increases the complexity of the set-up.

On a final note, we envisage that this chemistry technology can be scalable to large-scale carbonylation reactions. However, to achieve this goal a flow process for the $CO_2$ to CO would necessarily have to be developed to keep the reactor volumes to a convenient size. Although we believe this is achievable, such technologies would require a greater focus on instrument and reactor design. Finally, because of the ease in performing these electrochemical reductions with the described equipment, we foresee that the chemistry presented in this paper will have a significant impact on promoting others to apply electrochemical reactions in organic synthesis besides $CO_2$ reduction.

## Methods

**Synthesis of moclobemide (1)**. In a flame-dried two-chamber reactor (see Supplementary Fig. 14) charged with stirring bars were added Xantphos Pd G4 (24 mg, 5 mol%), 1-chloro-4-iodobenzene (112 mg, 0.5 mmol) DABCO (119 mg, 1.0 mmol), THF (3 ml) and 2-morpholinoethylamine (113 µl, 1.0 mmol) to chamber A. FeTPP (6 mg), TBABF$_4$ (1.10 g), DMF (35 ml), and TFE (2 ml) were introduced to chamber B. Electrodes were mounted and the glassware was sealed with screw caps fitted with Teflon-coated silicone seals. The solution in chamber B was bubbled through with $CO_2$ for 10–15 min to achieve saturation (outlet located in chamber A). The ElectroWare was set up using the galvanostatic configuration E1 (see Supplementary Fig. 18), electrodes were connected, and electrolysis was commenced while stirring the solution in both chambers. Chamber B was at room temperature while chamber A was placed in a preheated heat block.

Stirring was continued for 18 h, after which the chemical reaction was cooled to room temperature and the volatiles were removed in vacuo. The crude residue was purified by flash column chromatography to yield the desired product **1**. All yields are average of two runs. For NMR spectra and specific conditions, see Supplementary Methods.

**Data availability**. The data supporting this work are available from the authors upon reasonable request. The data that support the findings of this study are available within the paper and its Supplementary Information file or are available on request from the corresponding author upon request.

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

## Acknowledgements

We thank the Danish National Research Foundation (grant no. DNRF118) and Aarhus University for financial support.

## Author contributions

M.T.J., M.H.R. and A.K.R. contributed equally to this study. M.T.J., M.H.R., A.K.R., D.U.N., X.-M.H. and R.W.J. designed and performed the optimisation, equipment design and scope experiments. All authors analysed and interpreted the data from these experiments. S.U.P., T.S. and K.D. conceived and coordinated the study, and also wrote the paper with input or editing from all authors.

## Additional information

**Competing interests:** T.S. is co-owner of SyTracks a/s, which commercialises the COgen and SilaCOgen. The remaining authors declare no competing financial interests.

