## [Peer Review File · Nature Communications]

Reviewers' comments:

Reviewer #1 (Remarks to the Author):

This manuscript entitled "Scalable carbon dioxide electroreduction coupled to carbonylation chemistry" by Mikkel T. Jensen and co-workers describes an operationally simple and low-cost electrochemical set-up for the production of CO for carbonylation chemistry. Carbonylation chemistry is very important for organic synthesis, especially in pharmaceutical industry. The introduction of CO onto organic molecules is an indispensable process, allowing for converting substrates to aldehydes, ketones, carboxylic acids or related compounds. As CO gas is quite toxic, how to introduce CO gas to reaction systems in a safe way is an obvious challenging task. The set-up and design demonstrated in this manuscript provide a possibility that using Non-toxic substrate CO₂ as the starting material for carbonylation chemistry. However, this manuscript has following issues:

1. This work is not the first case of combination CO production with carbonylation chemistry, such as "Advanced Synthesis & Catalysis, 2015, 357(13): 2931-2938" has demonstrated a set up to produce CO from HCOOH by zeolites for palladium-catalyzed carbonylation reaction (Aminocarbonylation catalyzed by Xantphos Pd, similar to this manuscript), which is more safe, mild and efficient to introduce CO into molecules for organic synthesis (without CO₂ Cylinders and electrochemistry set up). Also the "Acc. Chem. Res. 49, 594-605 (2016)." published by one of the co-authors have demonstrated solid CO-releasing molecules and "Two-Chamber Reactors and Carbon Monoxide Precursors for Safe Carbonylation Reactions".

From aspect of laboratory synthesis, the design in this manuscript might increase the complexity of carbonylation experiments (such as CO₂ Cylinder and electrochemistry set up are required); from the point of industrial synthesis, the reaction system in this manuscript will obviously increase the cost. The author did not point out the advantages compared to the published work and methods, it's hard to persuade people to accept the design in this manuscript.

2. The CO₂ reduction system reported in this work are not original (Ref. 13-15); meanwhile, the "Two-Chamber Reactors" also have been reported (Ref. 21). In this work, the authors only combined two known systems in a physical way, I do not think this manuscript has represent an advance in understanding likely to influence thinking in the field, at the same time, the novelty of this work may not meet the demand of Nature communications.

Reviewer #2 (Remarks to the Author):

The manuscript entitled "Scalable carbon dioxide electroreduction coupled to carbonylation chemistry" by Skrydstrup et. al. demonstrates an elegant way of conversion of CO₂ (atmospheric!) to value added chemicals through multi step process. CO₂ capture from atmosphere, its heat release to a Feporphyrin containing chamber, its conversion to CO and its channel to another chamber where it takes up CO for Pd catalysed carbonylation reaction leading to useful chemicals is the theme of the manuscript. This is a very timely contribution on a very contemporary subject. While each step is well documented in literature, novelty of this work lies in their suitable combination. Thus this is recommended for publication in Nature communication after revisions.

Some specific points are:

1. In page 2, the authors claimed that stainless steel electrode revealed faster kinetics than iron electrodes. Are they pointing towards electron transfer kinetics? If so (likely it is!), Supp Fig. 6 is not sufficient. Probably the authors should try finding out Tafel slope for each of the electrodes.

2. In page 2, the authors also stated that the catalyst concentration is pivotal for controlling the chemoselectivity of CO₂ reduction. Please justify.

3. The authors used TFE instead of PhOH as source of weak acid for CO₂ reduction for their operation. Is there any specific reason for that? Please clarify.

Some minor comments are:

1. The authors should try doing the CO₂ reduction in water (several works are going on now with Fe porphyrin as catalyst) instead of using DMF as solvent. This would enhance the impact of the manuscript.

2. The paper could be written in more compact way.

Reviewer #3 (Remarks to the Author):

Daasbjerg, Skrydstrup and coworkers disclosed, in this manuscript, an interesting set up for carrying out sequential reactions for the synthesis of bioactive molecules. They ingeniously combined selective electroreduction of carbon dioxide to CO with efficient palladium-catalyzed carbonylation reactions, in which the amount of carbon monoxide was entirely controllable, avoiding the direct usage of CO. From synthetic point, the current transformation is very attractive and it might inspire the broad interest on the further explore novel reactions based on the current systems. Overall, this manuscript is quite interesting and will attract broad interest from the readership of Nature Comm. Thus, I recommend its publication after minor revision:

1) The reference electrode should be mentioned when the applied potential appears in the manuscript. It didn't make sense to only include the value.

2) For Figure 2c and 3b, the applied potential for the reduction of CO₂ should be -4.3V vs Ag/AgCl.

3) The reviewer was confused with the anode (counter) electrode, is it same as cathode or using other material? Normally, it will consume the cathode electrode when doing reduction reactions.

4) The reviewer suggests the author include some objective comments on the limitations and shortcomings of this system for the fully understanding by the readers.

The main text is well written and supporting information appears to be complete with all necessary materials. The procedural details and references are also appropriate.

Reviewers' comments:

Reviewer #1 (Remarks to the Author):

This manuscript entitled "Scalable carbon dioxide electroreduction coupled to carbonylation chemistry" by Mikkel T. Jensen and co-workers describes an operationally simple and low-cost electrochemical set-up for the production of CO for carbonylation chemistry. Carbonylation chemistry is very important for organic synthesis, especially in pharmaceutical industry. The introduction of CO onto organic molecules is an indispensable process, allowing for converting substrates to aldehydes, ketones, carboxylic acids or related compounds. As CO gas is quite toxic, how to introduce CO gas to reaction systems in a safe way is an obvious challenging task. The set-up and design demonstrated in this manuscript provide a possibility that using Non-toxic substrate CO₂ as the starting material for carbonylation chemistry. However, this manuscript has following issues:

1. This work is not the first case of combination CO production with carbonylation chemistry, such as "Advanced Synthesis & Catalysis, 2015, 357(13): 2931-2938" has demonstrated a set up to produce CO from HCOOH by zeolites for palladium-catalyzed carbonylation reaction (Aminocarbonylation catalyzed by Xantphos Pd, similar to this manuscript), which is more safe, mild and efficient to introduce CO into molecules for organic synthesis (without CO₂ Cylinders and electrochemistry set up). Also the "Acc. Chem. Res. 49, 594-605 (2016)." published by one of the co-authors have demonstrated solid CO-releasing molecules and "Two-Chamber Reactors and Carbon Monoxide Precursors for Safe Carbonylation Reactions".

From aspect of laboratory synthesis, the design in this manuscript might increase the complexity of carbonylation experiments (such as CO₂ Cylinder and electrochemistry set up are required); from the point of industrial synthesis, the reaction system in this manuscript will obviously increase the cost. The author did not point out the advantages compared to the published work and methods, it's hard to persuade people to accept the design in this manuscript.

Response: The reviewer states a good point! There are several works already showing the combination of CO production with carbonylation chemistry. However, compare with the published work we believe the current report features a new way of approaching the problem employing CO₂ as a starting material which in itself is an extremely interesting C₁ feedstock since it in opposition to the both the COgen and the formic acid is not derived from petrochemical industry. CO₂, as a greenhouse gas, causes a series of environmental issues, and currently the atmospheric CO₂ concentration is increasing. Therefore, increasing efforts has been made to convert CO₂ into various chemicals, of which the electrochemical reduction of CO₂ to CO represents an important pathway for CO₂ valorisation. Though CO₂ has been

successfully converted to CO, the facile utilisation of the CO₂-derived CO has not been studied. From this perspective, this work represents an advance in the area of CO₂ valorisation.

2. The CO₂ reduction system reported in this work are not original (Ref. 13-15); meanwhile, the “Two-Chamber Reactors” also have been reported (Ref. 21). In this work, the authors only combined two known systems in a physical way, I do not think this manuscript has represent an advance in understanding likely to influence thinking in the field, at the same time, the novelty of this work may not meet the demand of Nature communications.

Response: We agree with the reviewer that the CO₂ reduction system reported in this work are not original (Ref. 13-15) and the “Two-Chamber Reactors” also have been reported (Ref. 21). However, the core concept in this work of “coupling CO₂ electroreduction to CO with subsequent carbonylation reaction” is original and important. There are many reports converting CO₂ to CO via electroreduction. Apparently, CO is not the end product. However, there is no report up to date demonstrating the feasibility of using the CO₂-derived CO for further reactions to get more valuable products. From this perspective, this work represents an advance in the area of CO₂ valorisation.

Reviewer #2 (Remarks to the Author):

The manuscript entitled “Scalable carbon dioxide electroreduction coupled to carbonylation chemistry” by Skrydstrup et. al. demonstrates an elegant way of conversion of CO₂ (atmospheric!) to value added chemicals through multi step process. CO₂ capture from atmosphere, its heat release to a Feporphyrin containing chamber, its conversion to CO and its channel to another chamber where it takes up CO for Pd catalysed carbonylation reaction leading to useful chemicals is the theme of the manuscript. This is a very timely contribution on a very contemporary subject. While each step is well documented in literature, novelty of this work lies in their suitable combination. Thus this is recommended for publication in Nature communication after revisions.

Some specific points are:

1. In page 2, the authors claimed that stainless steel electrode revealed faster kinetics than iron electrodes. Are they pointing towards electron transfer kinetics? If so (likely it is!), Supp Fig. 6 is not sufficient. Probably the authors should try finding out Tafel slope for each of the electrodes.

Response: The reviewer’s point about comparison of the electrode kinetics of the iron vs the stainless steel electrodes purely on basis of a single electrolysis experiment is valid. As a consequence, we have chosen to remove the sentence from the manuscript since it is not a main

point of the manuscript to compare electrode materials. The main reason for choosing stainless steel as an electrode material over iron was to avoid the oxidation of the counter electrode observed when iron was used which reduced the lifetime of the electrodes and was a pollutant in the system.

2. In page 2, the authors also stated that the catalyst concentration is pivotal for controlling the chemoselectivity of CO₂ reduction. Please justify.

Response: The importance of sufficient catalyst concentration in order to obtain a good chemoselectivity for the CO₂ reduction can be explained by looking at the two main components of the current:

$$i_{cat} + i_{background} = i_{total}$$

*By increasing the catalyst loading we increase the i_{cat} component resulting in more CO production. It is well documented both from the supporting information in the section with "Control Experiments" (supplementary p 10) and literature (for instance: *Electrochim. Act.* 39, 1833-1839 (1994)) that background current mainly originates from reduction of protons (TFE or water) to hydrogen, which is further catalysed by the stainless steel electrode or other iron based electrodes.*

3. The authors used TFE instead of PhOH as source of weak acid for CO₂ reduction for their operation. Is there any specific reason for that? Please clarify.

*Response: Initially, PhOH was tested as a weak acid due to its widespread use in more recent literature. Unfortunately, we experienced a fast decrease in activity of the system in the one compartment cell due to oxidative polymerization of the PhOH and film formation on the counter electrode as described by Ferreira et.al (*Electrochim. Act.* 52, 434-442 (2006)). Hence, we choose to use TFE, which has also been proven to significantly enhance the catalytic activity of iron porphyrin. This is now explained in the text.*

Some minor comments are:

1. The authors should try doing the CO₂ reduction in water (several works are going on now with Fe porphyrin as catalyst) instead of using DMF as solvent. This would enhance the impact of the manuscript.

*Response: For the CO₂ reduction, a solvent such as water would definitely both decrease cost and be more environmental friendly. We are aware of several interesting systems applying iron or cobalt porphyrin in water, such as described in *Chem. Commun.* 52, 12084-12087 (2016), *J. Am. Chem. Soc.* 2016, 138, 2492-2495, *Science* 2015, 349, 1208-1213, *J. Am. Chem. Soc.* 2015, 137, 14129-14135, and *Angew. Chem. Int. Ed.* 2017, DOI: 10.1002/anie.201701104. Indeed, we also considered the possibility of coupling the CO₂ reduction in water with the carbonylation reaction. However, for the system described in this work, water based systems were avoided*

since water oxidation (at the counter electrode) would lead to O₂ formation towards which the Pd-based follow up chemistry is rather sensitive. Even the formation of tiny amount of O₂ will deactivate the catalysts for carbonylation and thus make the reaction inefficient. Of course, the O₂ problem can be solved by separating the counter electrode from the working electrode, but this would require a third compartment and makes the setup more complicated.

2. The paper could be written in more compact way.

Response: We have now converted the manuscript to the style of Nature Communications.

Reviewer #3 (Remarks to the Author):

Daasbjerg, Skrydstrup and coworkers disclosed, in this manuscript, an interesting set up for carrying out sequential reactions for the synthesis of bioactive molecules. They ingeniously combined selective electroreduction of carbon dioxide to CO with efficient palladium-catalyzed carbonylation reactions, in which the amount of carbon monoxide was entirely controllable, avoiding the direct usage of CO. From synthetic point, the current transformation is very attractive and it might inspire the broad interest on the further explore novel reactions based on the current systems. Overall, this manuscript is quite interesting and will attract broad interest from the readership of Nature Comm. Thus, I recommend its publication after minor revision:

1) The reference electrode should be mentioned when the applied potential appears in the manuscript. It didn't make sense to only include the value.

Response: We have clarified the potential reference issue in the text, in that for a two-electrode set-up a reference potential cannot be defined as it is usually done for a conventional three electrode set-up. In the former case the counter electrode cannot be seen as a reference electrode since the potential is not fixed for this electrode.

2) For Figure 2c and 3b, the applied potential for the reduction of CO₂ should be -4.3V vs Ag/AgCl.

Response: The potential here is again stated vs the counter electrode (see the comment above).

3) The reviewer was confused with the anode (counter) electrode, is it same as cathode or using other material? Normally, it will consume the cathode electrode when doing reduction reactions.

Response: The anode (counter) electrode is made of stainless steel 316 as the working electrode is. The referee is correct that many electrochemical procedures make use of a sacrificial anode. In the present system, it is not clear to which extent the stainless steel electrode may serve as a sacrificial electrode or if the solvent or supporting electrolyte rather get oxidised.

4) The reviewer suggests the author include some objective comments on the limitations and shortcomings of this system for the fully understanding by the readers.

Response: In principle, upscaling of the reported procedure to industrially relevant dimensions should be possible, although the use of rather large volumes of TBABF₄ and DMF in that instance would be unfortunate. The relatively large volume of DMF used was mainly necessary in order to dissolve sufficiently large amounts of CO₂ in the system, therefore an important challenge if this system should be scaled up would be to flow CO₂ into the system in order to reduce the amount of solvent and electrolyte. However, in that case new problems could arise after longer times concerning the oxidation process at the counter electrode, which in that case should be controlled more carefully.

For laboratory use one of the main shortcomings when comparing to other known systems such as the COgen previously reported by Skrydstrup et.al. is the fact that the CO release rate is not instant. In contrast, the electrochemical approach allows the CO formation rate to be accurately controlled, thereby avoiding the large buildup of CO which, potentially, could cause a poisoning of the catalyst

The main text is well written and supporting information appears to be complete with all necessary materials. The procedural details and references are also appropriate.

Response: Thank you.

REVIEWERS' COMMENTS:

Reviewer #2 (Remarks to the Author):

The authors have addressed the comments by this reviewer satisfactorily. This paper is now acceptable.

Reviewer #3 (Remarks to the Author):

The revised manuscript has been improved, and most of questions have been well addressed. Thus, this referee believes that the revised manuscript should meet the standard of Nature Communication.